# A metasurface-based full-color circular auto-focusing Airy beam transmitter for stable high-speed underwater wireless optical communications

Junhui Hu [1,4], Zeyuan Guo[2,4], Jianyang Shi [1], Xiong Jiang[2], Qinmiao Chen[2], Hui Chen[3], Zhixue He[3], Qinghai Song [2,3], Shumin Xiao [2,3] ✉, Shaohua Yu [1,3], Nan Chi [1,3] ✉ & Chao Shen [1,3] ✉

Due to its unique intensity distribution, self-acceleration, and beam self-healing properties, Airy beam holds great potential for optical wireless communications in challenging channels, such as underwater environments. As a vital part of 6G wireless network, the Internet of Underwater Things requires high-stability, low-latency, and high-capacity underwater wireless optical communication (UWOC). Currently, the primary challenge of UWOC lies in the prevalent time-varying and complex channel characteristics. Conventional blue Gaussian beam-based systems face difficulties in underwater randomly perturbed links. In this work, we report a full-color circular auto-focusing Airy beams metasurface transmitter for reliable, large-capacity and long-distance UWOC links. The metasurface is designed to exhibits high polarization conversion efficiency over a wide band (440-640 nm), enabling an increased data transmission rate of 91% and reliable 4 K video transmission in wavelength division multiplexing (WDM) based UWOC data link. The successful application of this metasurface in challenging UWOC links establishes a foundation for underwater interconnection scenarios in 6G communication.

Since a non-diffracting curved-propagating Airy beams conforming to the Helmholtz equation was introduced in optics[1], they have exhibited significant potentials in particle manipulation[2,3], high-resolution microscopy[4–6], and optical bullets[7], due to its unique self-healing, non-diffractive and self-accelerating properties. Airy beams can be generated by spatial light modulators[8,9], optical lens systems[10,11], non-linear methods[12,13]. However, the size of the pixel affects the generation of Airy beams, and the optical systems required by these methods are bulky and cannot be integrated with photonic chips.

Metasurfaces are ultrathin and integrable two-dimensional planar artificial materials composed of subwavelength metallic or dielectric unit structures with the ability to tune the amplitude, phase, and polarization properties of light. Compared to traditional phase plates, metasurfaces have smaller pixel size and larger diffraction angles, enabling to achieve Fourier transformation in short range. These advantages make them of great research value in ultra-thin lenses[14,15], holographic imagings[16,17], structural colors[18,19] and integrated photonics[20–22]. Recently, generation of visible Airy beams by using

[1]Key Laboratory for Information Science of Electromagnetic Waves (MoE), School of Information Science and Technology, Fudan University, Shanghai, China. [2]Ministry of Industry and Information Technology Key Lab of Micro-Nano Optoelectronic Information System, Guangdong Provincial Key Laboratory of Semiconductor Optoelectronic Materials and Intelligent Photonic Systems, Harbin Institute of Technology, Shenzhen, China. [3]Peng Cheng Laboratory, Shenzhen, China. [4]These authors contributed equally: Junhui Hu, Zeyuan Guo. ✉e-mail: shumin.xiao@hit.edu.cn; nanchi@fudan.edu.cn; chaoshen@fudan.edu.cn

metasurface has been studied toward broad-band, full-color and high-efficiency. Employing synthetic-phase method contributes to producing tunable focal length red Airy beam[23]. Fabricating various meta-cells can be utilized to collectively cover the 450–600 nm response band[24]. However, it is still challenging to achieve a high conversion efficiency across the visible color regime using a single structure. In optical communication and imaging systems, a uniform spectral response in optical devices is imperative, as it eliminates the need for additional power distribution or filters.

In the realm of optical communication, Airy beams have yet to showcase their full potential, while orbital angular momentum (OAM) has found widespread application in mode division multi-plexing communication systems[25–27] and adaptive optical compensation[28–30]. Airy beams are mainly employed for information or image transmission bypassing obstacles due to their self-accelerating property[31–33]. Underwater wireless optical communication (UOWC) is considered a core scenario for 6G wireless network[34,35]. The wavelength selection in the presence of complex channel disturbances poses significant challenges to achieving compatibility and reliability in high-speed communication. The conventional blue Gaussian beam-based UWOC systems are strongly affected by time-varying conditions, the presence of underwater obstacles and air bubbles deteriorates the data rate and its stability. Simultaneously, OAM or Bessel beams based on spatial light modulators have not demonstrated significant advantages in mitigating the impact of underwater bubbles[36]. In Contrast, the utilization of visible-band Airy beams in UWOC holds promising potential for establishing more reliable links. Theoretically, Airy beams, with their self-healing property and distinctive main lobe and side lobe distributions, are expected to exhibit greater resilience to disturbances in underwater channels compared to Gaussian beams. Nevertheless,

the current lack of practical experiments hinders the confirmation of this perspective.

In this work, we propose an adaptive UWOC link, which shows reliable, large-capacity, long distance advantages over conventional system based on Gaussian beam. System relies on multi-wavelength lasers[37] and an ultra-broadband metasurface for visible-band Gaussian to circular auto-focusing Airy beam (CAFAB) conversion. A high-efficient CAFAB metasurface with uniform conversion efficiency across the entire visible light band is designed, demonstrating its beam pro-pagation characteristics and occlusion simulation. Then, the performance of red, green and blue (RGB) CAFAB lasers employing our metasurface in UWOC application scenarios is evaluated. The feasi-bility of Airy beam in addressing the impact by obstacles and air bubbles in complex underwater environment is discussed. It is indi-cated that Airy beam can alleviate the degradation of received optical power (ROP) caused by disturbance introduced by objective occlusion and air bubbles. Also, the benefit of ultra-broadband is reflected in the flexible adjustment of wavelengths for different water qualities and WDM scheme. Meanwhile, we show a robust and error-free 4 K ultra-high-resolution video transmission in the challenging underwater channel by using the full-color metasurface. Our work facilitates the development of high-efficiency ultra-broadband metasurface in visible regime for CAFAB generation. And the RGB Airy beams based reliable high-speed and long-range UWOC system is demonstrated.

## Results

### Adaptive UOWC links via a multi-wavelength Gaussian-to-CAFAB conversion metasurface

Figure 1 presents the framework and performance comparison between traditional UWOC link and our proposed adaptive link (Fig. 1a). In traditional links (Fig. 1b), blue (~450 nm) band is

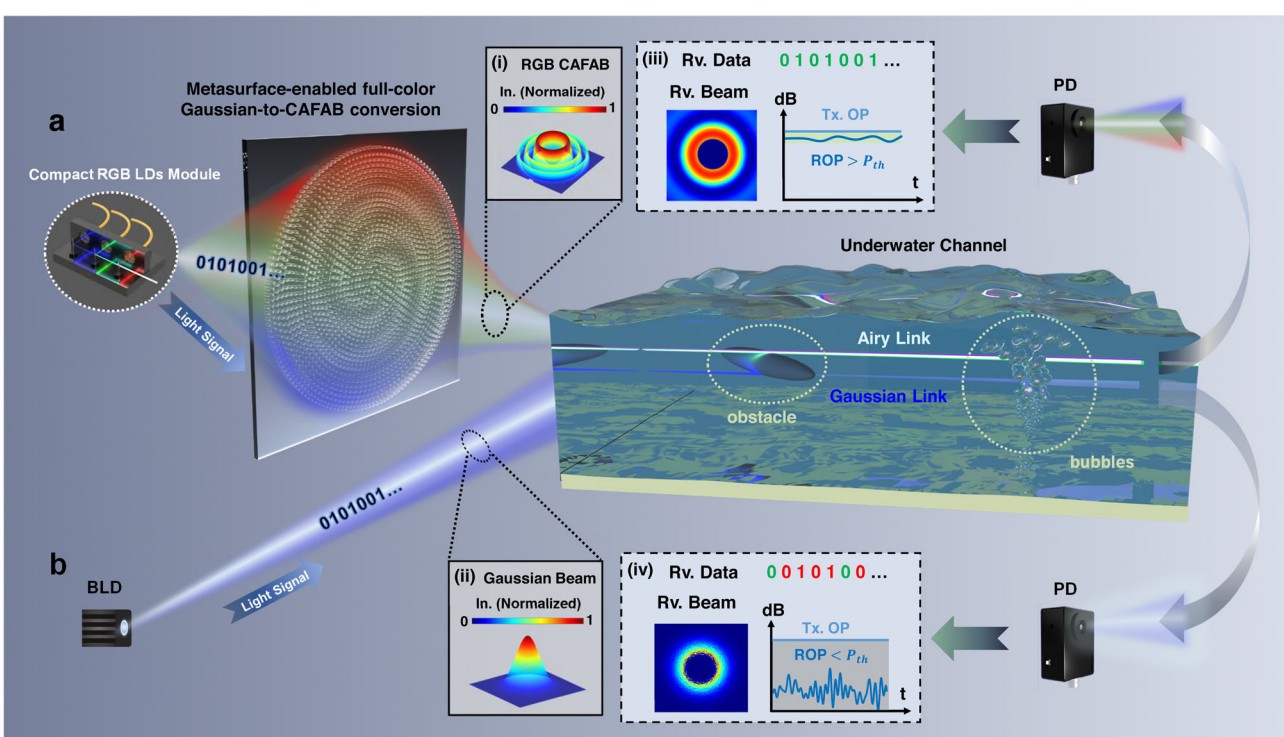

**Fig. 1 | Schematics of conventional Gaussian beam based UWOC system and our adaptive link with multi-wavelength Airy sources transformed by an ultra-broadband metasurface platform. a** Tri-color Gaussian beams are generated by a compact 6 × 4.4 × 3 cm³ LDs module. The conversion of Gaussian to circular Airy light is accomplished by a high-efficient metasurface covering the visible band. And multiple wavelengths are used to achieve WDM scheme. Inset (i), (ii) represent the intensity profile of CAFAB and Gaussian beam, respectively. In. stands for intensity. Insert (iii), (iv) represent received information of link a and link b, respectively. Rv. stands for received. **b** traditional UWOC links usually use blue Gaussian beam as the emission light source. Complex underwater environment limits the receiving per-formance of the system, including beam quality and bit error rate (BER).

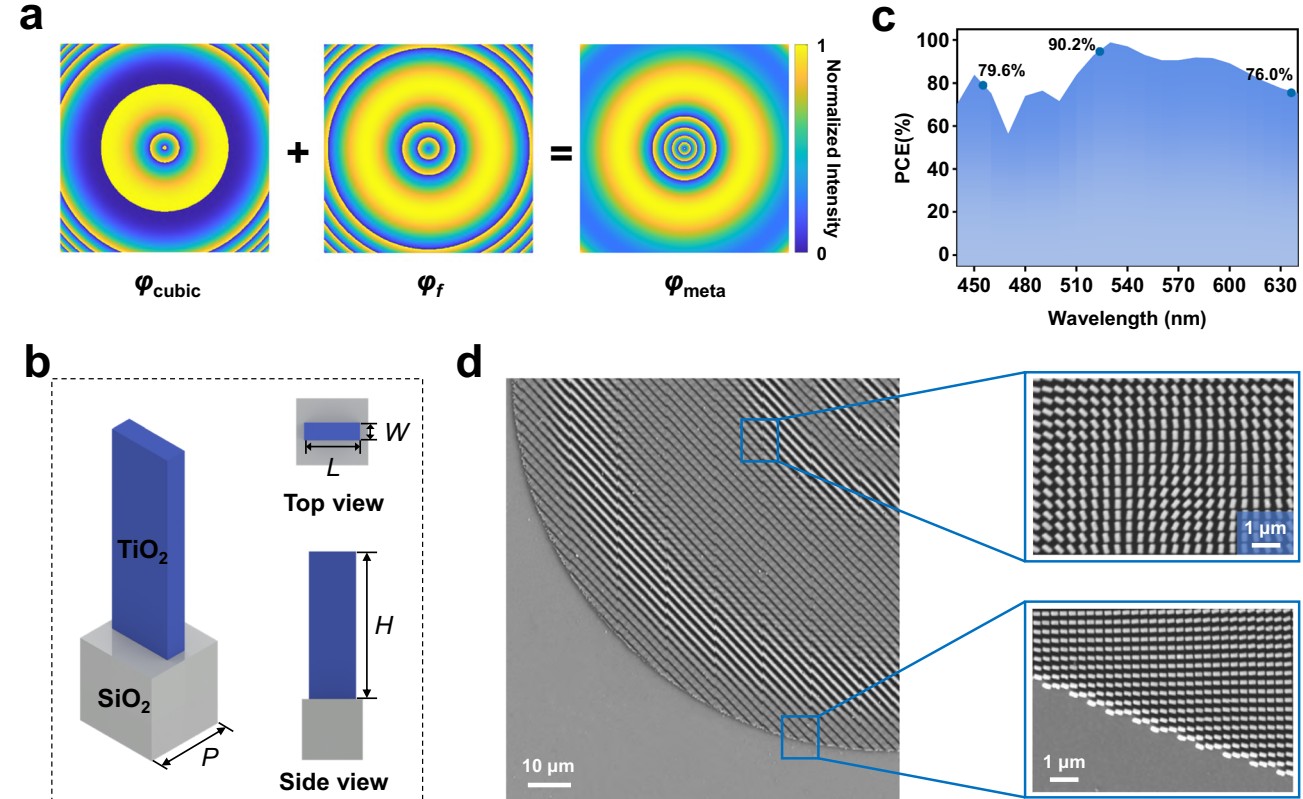

**Fig. 2 | Ultra-broadband CAFAB metasurface design and its performance.** **a** Phase distribution of superposition of cubic phase and lens phase. **b** Left: schematic diagram of the meta, right: top view and side view of the meta. **c** The polarization conversion efficiency (PCE) of the metasurface when the incident wavelength is 440 nm–640 nm. **d** Scanning electron microscopy (SEM) image of the details of a fabricated polarization-dependent metasurface.

considered to be the best communication wavelength for UOWC due to its smallest attenuation coefficient of 0.007 m$^{-1}$ under pure water. However, in turbid water with impurity concentrations higher than 8 g/m$^3$, the attenuation coefficient of green light is the smallest. The advantage of red light is reflected in the high-speed data rate within short distance (see Fig. S1 in the Supporting Information). Meanwhile, experiments have shown that red light has a greater attenuation length in waters with high chlorophyll content[38] (Offshore waters, etc.). Therefore, the monochromatic blue light based UOWC system may has limited application scenarios. Figure 1(ii) shows the Gaussian beams intensity distribution. When the obstacles present in the underwater channel, there will be a significant optical loss, leading to a massively reduced ROP. Also, air-water turbulence due to the presence of bubbles in UWOC link causes severe disturbances in the ROP, as shown in Fig. 1(iv). When the ROP level is lower than the threshold power ($P_{th}$, means minimum ROP for efficient bit-power loaded signal transmission), BER will exceed the forward error correction (FEC) threshold.

In order to support high-reliability, large-capacity and long-distance UWOC system, an adaptive scheme using tri-color Airy beams is proposed (Fig. 1a). At transmitting end, the Triser module[37] combines collimated RGB laser beams. Using such transmitter, we could adaptively select the optimal wavelength according to different water qualities, as well as increase the transmission capacity by WDM scheme. RGB CAFAB are generated by employing our designed ultra-broadband metasurface, the intensity distribution of the beam is illustrated in Fig. 1(i). High conversion efficiency throughout the passband prevents large loss of optical power. In the region of self-focusing, the obstruction resulting from small-scale obstacles is mitigated by the energy carried in the side lobes. Within the divergent region, the system's adaptability to obstacles is improved due to the distinctive circular energy distribution of Airy beam. In the presence of air bubbles, the intensity distribution of the Airy beam exhibits robustness against perturbations[39]. The received beam and data signals are shown in Fig. 1(iii). The photoelectric detector with limited aperture can receive extremely stable ROP as much as possible. The ROP is above threshold $P_{th}$ which means that the original binary code stream will be perfectly recovered through the DSP.

## Principle and performance of ultra-broadband Airy beam metasurface

Figure 2a shows the phase distribution model of the metasurface based CAFAB. The phase distribution of the metasurface in polar coordinates can be obtained by superimposing the cubic phase (Fourier transformation of the Airy function) and the Fresnel lens phase, which can be expressed as:

$$\varphi_{meta}(r) = \frac{(2\pi b_1)^3 (r - r_0)^3}{3} - \frac{2\pi \left[ \sqrt{(r - r_0)^2 + f^2} - f \right]}{\lambda} \quad (1)$$

where $f$ is the focal length of the Fresnel lens, $r$ is the coordinate with the center of the metasurface as the origin, $b_1$ is the attenuation coefficient, $r_0$ is the main lobe radius of the Airy beam, and $\lambda$ is the wavelength of the incident light. (See Section 1 in the Supporting Information for detail). The constructed metasurface is shown in Fig. 2b. The basic cell of the metasurface is composed of SiO$_2$ with a period of $P = 300$ nm as the substrate and stacked TiO$_2$ nanopillars with a height of $H = 800$ nm. According to the PB phase principle, when the incident beam is circularly polarized light, the rotation angle of the meta is twice the designed phase, and the meta is equivalent to a half-

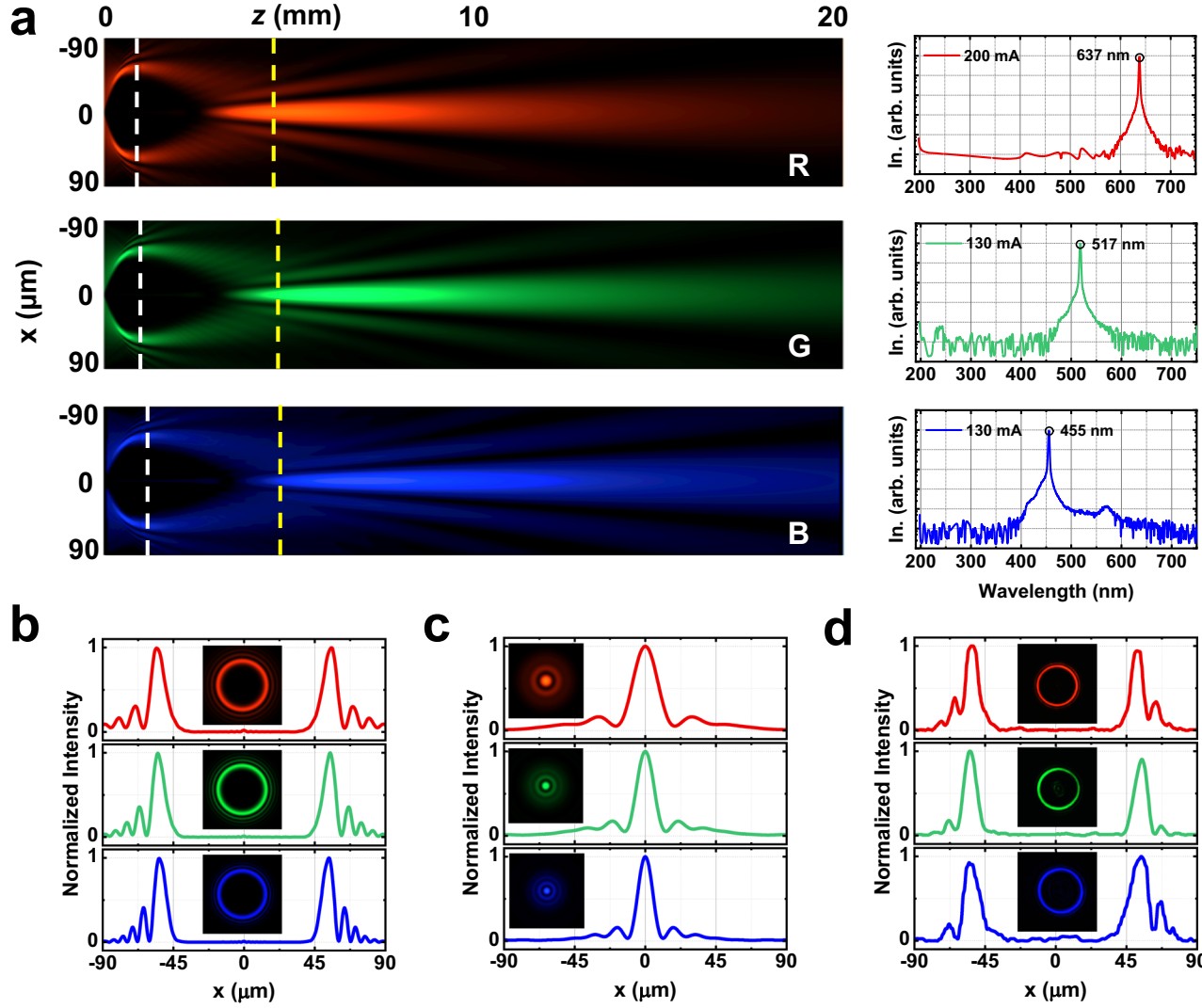

**Fig. 3 | CAFAB simulation and measurement for the wavelengths of 455, 517 and 637 nm. a** Left part is the x–z section of the Airy beam propagation trajectory simulation with the $z = 0$ plane as the metasurface plane, right part are the spectra of RGB lasers in log scale. In. stands for intensity. Simulation of x–y RGB Airy field distribution (**b**) at position $z = f$ (white dash), (**c**) at the focus position $z = 4.6$ mm, 4.7 mm, 4.8 mm, respectively (yellow dash). **d** Experimental measurement of RGB Airy field distribution at the position $z = f$.

wave plate, which can change the polarization of the beam and convert the incident light. Therefore, optimizing the PCE of a single cell is crucial for the generation efficiency of Airy beams. As shown in Fig. 2c, we choose the length and width to be $L = 90$ nm and $W = 260$ nm for design, and the theoretical efficiencies at three wavelengths of 455 nm, 517 nm and 637 nm are 79.6%, 90.2% and 76.0% respectively. Figure 2d shows the SEM image of the manufactured Airy beam metasurface generator, which indicates the metasurface we prepared has a high verticality and uniformity.

To verify the effectiveness of the design process, we prepare and numerically demonstrate a metasurface that can generate RGB circular self-focusing Airy beams. Considering the computational constraints of the simulation work, the $r_0$, $b_1$ and focal length $f$ are set to 55 μm, 0.01 μm$^{-1}$ and 1 mm, respectively. The simulated light intensity distribution along the $x$-$z$ plane is shown in Fig. 3a. The wavelength of the RGB laser is measured as 637 nm, 517 nm and 455 nm at the optimal operating points of 200 mA, 130 mA and 130 mA. Taking the metasurface plane as the position of $z = 0$, the $z$-position of white dotted line is the focal length of the Fresnel lens, which indicates the position where the circular Airy beam is generated (Fig. 3b). With the origin as the center, the beam intensity

distribution in each direction conforms to the Airy function. The main lobe positions and the full widths at half maximum of the RGB beams are 55.6 μm, 54.9 μm, 54.8 μm and 10.2 μm, 8.7 μm, 8.4 μm, respectively. The yellow dotted line is the focal position of the CAFAB. The focal points of the beams all have a circular symmetrical shape, and the cross-section presents an Airy disk distribution, as shown in Fig. 3c.

CAFAB metasurface was fabricated according to the phase profile design of the synthetic metasurface. The phase profile consists of the cubic phase ($b_1 = 0.01$ μm$^{-1}$), the Fresnel holographic lens phase (focal length $f = 1$ mm). Measured actual RGB PCE are 63.55%, 77.5% and 63.10%, respectively. Optical measurements are shown in Fig. S4. Figure 3d shows the transverse light field distribution of the CAFAB generated at $z = f$. Since the single pixel of the metasurface is much smaller than that of the spatial light modulator, the beam generated has a higher resolution. The field profiles extracted from the white dashed lines in Fig. 3a. The position and full widths at half-maximum of the main lobe of the beam at the RGB wavelengths are 52.6 μm, 54.7 μm, 54.8 μm and 8.8 μm, 9.3 μm, 12.1 μm, respectively. The experimental results agree with the simulation results.

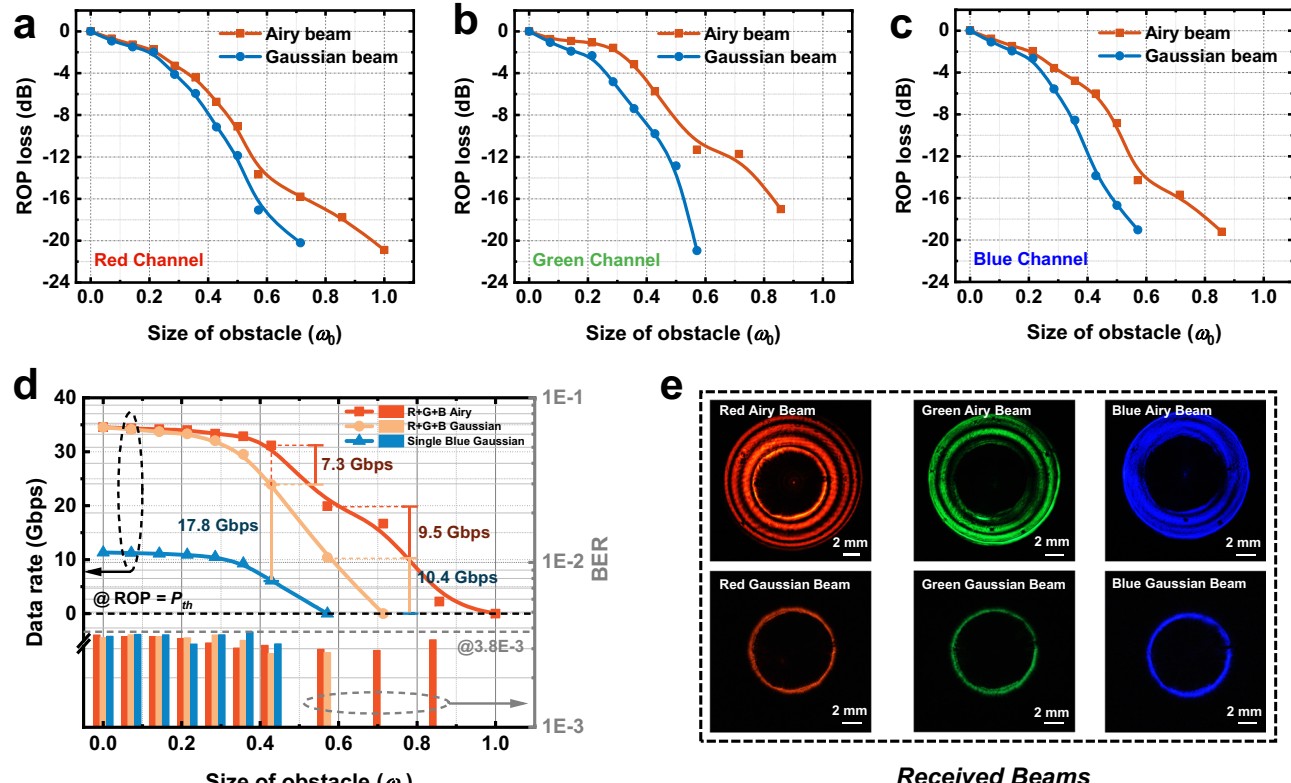

**Fig. 4 | Experimental results of the occlusion impact on UWOC systems.** ROP loss of Airy and Gaussian based red (**a**), green (**b**), and blue (**c**) channel versus the size of obstacle. The $\omega_0$ is normalized to maximum transmitter beam apertures of -14 mm. **d** Data rate and BER versus obstacle sizes. The emerging scheme with metasurface (R + G + B Airy beam), without metasurface (R + G + B Gaussian beam) and conventional scheme (single blue Gaussian beam) is compared. **e** Received optical beams after being blocked by an obstacle of 0.57 $\omega_0$ (-8 mm). The three receiving beams in first row represent RGB Airy beams with the metasurface, and figures in second row represent RGB Gaussian beam without the metasurface.

## Performance of the ultra-broadband metasurface based adaptive UWOC system

We built a UWOC experimental setup (see Fig. S4 in the Supporting Information) to further verify the feasibility of the adaptive UWOC scheme and show its communication performance. Figure 4a–c shows the measured ROP loss of the received RGB Airy and Gaussian beams under different occlusions. Here, the reference zero value is the ROP of the RGB Gaussian beam without obstacle, which are −0.68 dBm, −0.54 dBm, and 5.89 dBm respectively. The results indicate that RGB Airy beams can significantly improve the ROP loss of Gaussian beams with the increase of the obstacle size (The details are summarized in Table S1). In Fig. 4d, RGB Airy beams, RGB Gaussian beams and single blue Gaussian beam based UWOC systems are compared to deal with underwater obstacles of different scales. When the diameter of obstacle reaches 0.43 $\omega_0$, our proposed scheme is higher than RGB Gaussian beams transmission without metasurface by 7.3 Gbps. The monochrome blue Gaussian beam-based link is lower than the latter 17.8 Gbps. The BERs close to $3.8 \times 10^{-3}$ reveal system data rate approach the Shannon limit. And bit-power allocation schemes for RGB channels are demonstrated in the Supporting Information Fig. S5. Meanwhile, the ROP loss of RGB channels with metasurface are −6.73 dB, −5.73 dB and −6.01 dB, compared with the tri-channels without metasurface are reduced by 2.41 dB, 4.07 dB and 7.85 dB, respectively (see Table S1 in the Supporting Information). Then data rate of RGB Airy beams is degraded to 19.9 Gbps at 0.57 $\omega_0$ obstacle, which still exceeds 9.5 Gbps of RGB Gaussian beams. The ROP loss of single blue Gaussian beam is −19.03 dB, causing ROP lower than $P_{th}$ cannot support the effective bit-power loading data rate. Since the Airy beams has a respectively large residual energy in the side lobe, its

power loss with the central occlusion is not as severe as Gaussian beams (Fig. 4e). Normalized received beams indicate red channel has optimal Airy beam shape quality, saving 3.4 dB of ROP for the Gaussian loss. Green and blue channels have the ROP enhancement of 9.66 dB and 4.74 dB with metasurface. Further, RGB Airy beams can maintain data transmission within obstacle size of 0–1 $\omega_0$, while RGB Gaussian beams-based link will no longer be established when the diameter of occlusion reaches 0.71 $\omega_0$. These verify the great potential of Airy beams in UWOC transmission under the impact of obstacles.

Figure 5a illustrates the influence of air bubbles on the ROP of RGB Airy and Gaussian beams at different air pump flow rates. The normalized ROP (NROP) reference zero is set as the average value of the RGB Gaussian channels. At a $V_{pump}$ of 1 L/min, the striped heat map indicates that the power fluctuations caused by small bubbles are not severe. The average NROP values for the RGB Airy channels are measured as 0.17, 0.35, and 0.06 dB, respectively. Inclusion of the metasurface reduces the NROP fluctuation range by 0.24, 0.20, and 0.46 dB, respectively. Increasing $V_{pump}$ to 6 L/min significantly expands the fluctuation range of NROP to over 3 dB. However, the advantage of Airy beams remains evident as the standard deviation of the RGB channel NROP is reduced by 0.051, 0.097, and 0.20 when compared to Gaussian beams. Further increasing $V_{pump}$ to the maximum of 10 L/min results in a drastic degradation of the RGB Gaussian NROP response, with ranges of 20.83, 14.57, and 17.23 dB observed. Utilizing the metasurface, the fluctuation ranges of the tri-channel NROP can be reduced by 52.44%, 30.55%, and 27.96%, respectively.

Moreover, in order to assess the impact of different bubble scenarios on the effective transmission data rate distribution (as depicted in Fig. 5b), the amount of data per frame is reduced to increase the

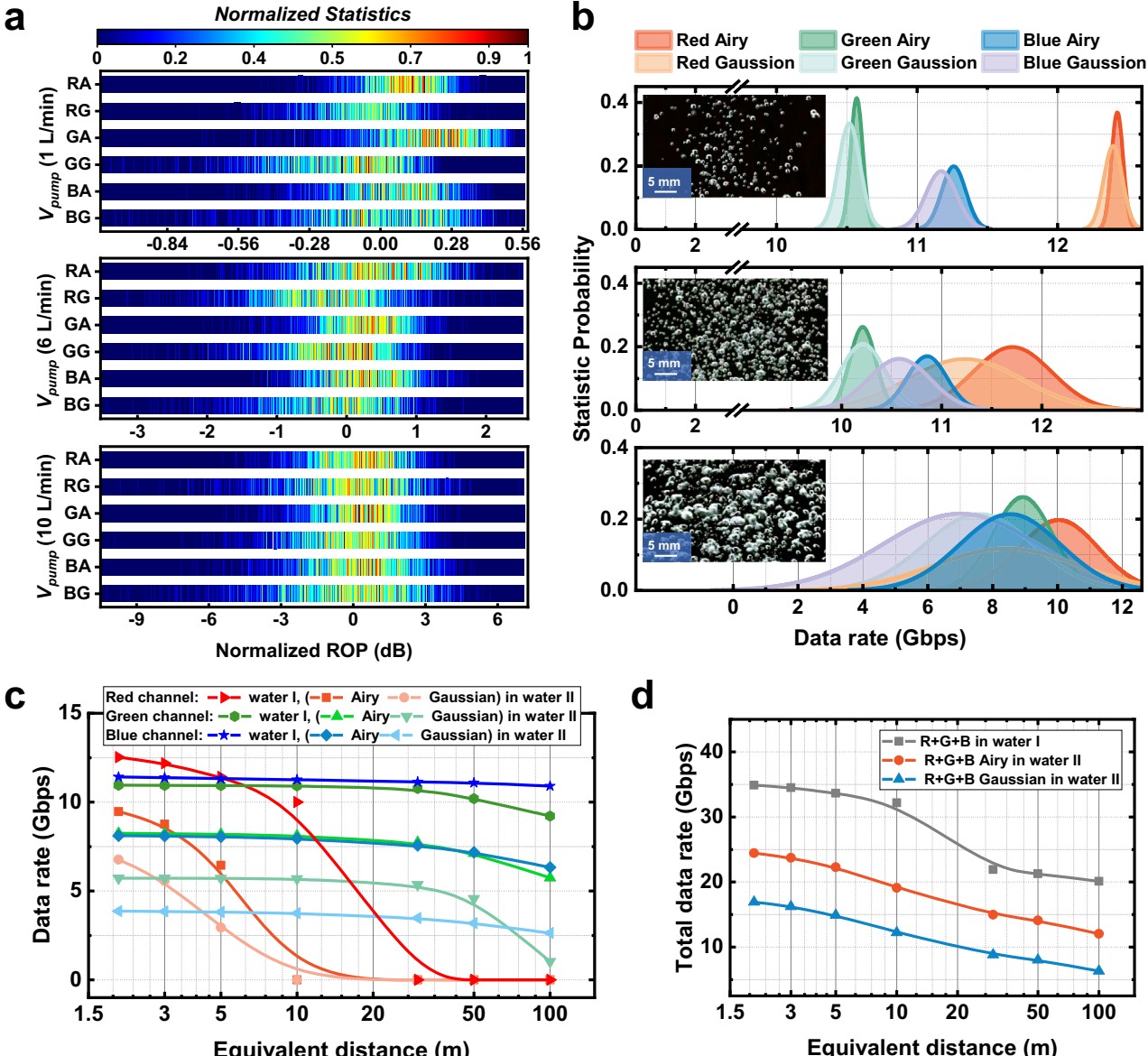

**Fig. 5 | Experimental results of the UWOC systems response to air bubbles and transmission distance. a** Normalized ROP statistics of red Airy (RA), red Gaussian (RG), green Airy (GA), green Gaussian (GG), blue Airy (BA) and blue Gaussian (BG) beams-based channel link. Different bubble conditions of 1 L/min, 6 L/min and 10 L/ min air pump flow rate are covered respectively. **b** Statistic probability of data rate for three bubble sizes. **c** Data rate of the RGB Airy and Gaussian beams-based channels versus the equivalent transmission distance when facing water I (pure water) and water II (0.43 $\omega_0$ obstacle with middle bubbles). **d** Total RGB data rate versus the distance under water I and water II.

number of receiving synchronization peaks in once sample. This adjustment facilitates the evaluation process. As the bit-power loading data rate is positively correlated with the signal-to-noise ratio, which is dependent on the level of ROP. The data rate distributions are fitted as Gaussian-like distributions (refer to Fig. S5 in the Supporting Information). The insets in the figure correspond to bubble images at three different air pump flow rates, as presented in Fig. 5b, and the estimated average bubble areas are 0.89, 1.63, and 2.55 mm², respectively. Results demonstrate the transmission of Airy beams exhibits a more stable data rate distribution when compared to Gaussian beams in the presence of interference from bubbles of varying scales.

Figure 5c shows the long-range performance of RGB Airy/Gaussian channels in pure water (water I) and water II consist of 0.43 $\omega_0$ obstacle and middle bubbles. To simulate the effect of long-range, an absorptive neutral density filter is employed, and the equivalent attenuation is calculated using the spectral attenuation coefficient.

The results reveal that red channel shows a notable advantage in short-distance transmission, primarily due to the prevalence of scattering disturbances. However, as the distance increases, the dominant factor becomes water absorption, resulting in a rapid decay of the data rate for red channel. And blue or green channel takes the lead. Importantly, the results shown in Fig. 5d demonstrate that the RGB Airy scheme effectively mitigates the degradation of data rates caused by simultaneous perturbations at different transmission distances. At a transmission distance of 100 m, the achievable data rate in the UWOC link under water I channel is 20.1 Gbps. By employing the metasurface-based full-color circular auto-focusing Airy beam transmitter, the data rate of the UWOC link under water II channel has been increased from 6.32 Gbps to 12.1 Gbps. Such significant improvement of 91% in achievable data transmission rate confirms the potential of our demonstrated adaptive UWOC system for high-speed and long-distance data link in challenging environments.

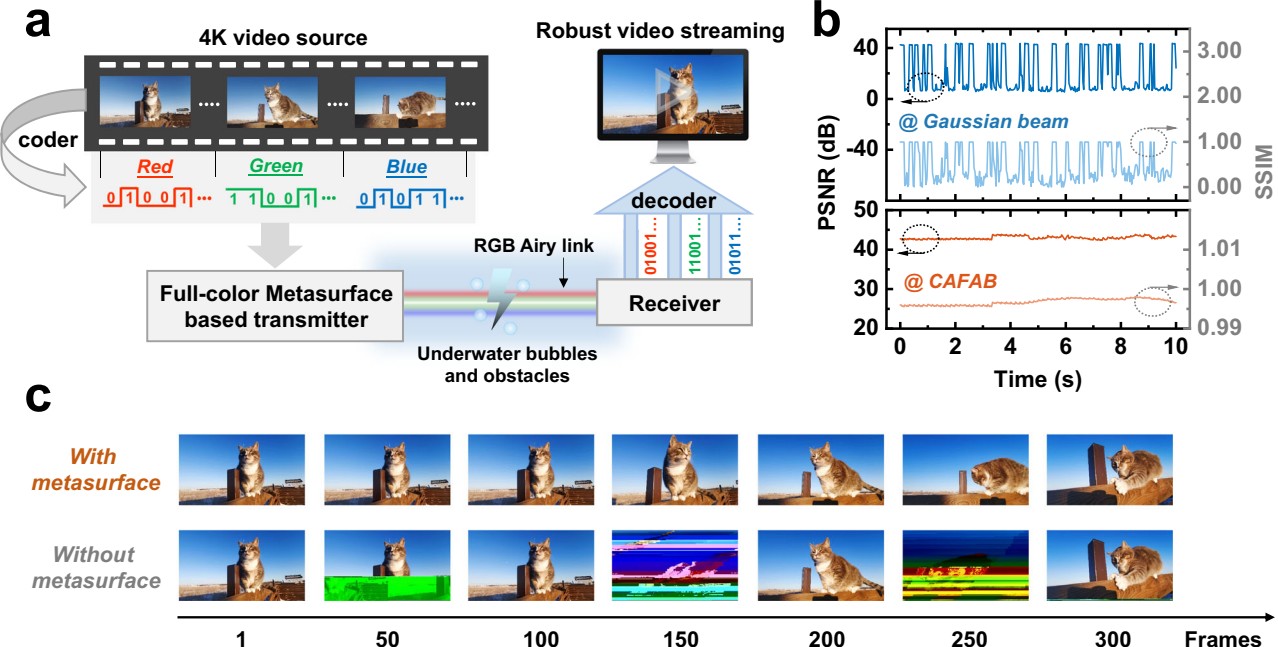

**Fig. 6 | Underwater 4 K video streaming using the full-color metasurface.**
**a** Schematic of the system. **b** The quality metrics, PSNR and SSIM of each frame received by the CAFAB (with metasurface) or Gaussian beam (without metasurface) based UWOC system versus time. **c** The received video frames with and without the full-color metasurface at different frame sequences.

Figure 6 present a 4 K ultra-high-resolution video transmission based on our high-speed full-color metasurface laser transmitter in the UWOC data link. The experimental results clearly present the enhancement of robustness against underwater bubbles and obstacles by using the metasurface. The system structure is illustrated in Fig. 6a. The 4 K video source is transformed into a transmittable bitstream signal through frame encoding. WDM technology enables high-speed transmission of the bitstream through RGB channels. Beam shaping into RGB CAFAB implemented by full-color metasurface significantly enhances the stability of the communication link in water II. This allows the receiver to decode the 4 K video data. Figure 6b shows a comparison of the quality metrics for 4 K video received by CAFAB (with metasurface) and Gaussian beam (without metasurface) based systems. Results significantly demonstrate the superior robustness of the CAFAB based system. The peak signal-to-noise ratio (PSNR) of each frame of the 4 K video remains around 43 dB, and the structural similarity (SSIM) is consistently above 99.3%. While in the Gaussian beam based UWOC link, the system can only recover parts of the frame images, as shown in Fig. 6c. The corresponding average PSNR and SSIM are 19.62 dB and 49.02%, respectively. The results suggest that significant attenuation from obstacles and larger power fluctuations caused by bubbles was presented in the system without metasurface. The advances of the CAFAB based system pave the path for robust, high-capacity UWOC links.

## Discussions

Here, we successfully fabricate an ultra-broadband metasurface for CAFAB generation, covering the 440–640 nm visible spectrum. The results presented in Fig. 2 demonstrate the metasurface exhibits high PCE at RGB regime, using a single meat cell. Conventionally, the generation of multi-wavelength Airy beams in visible-band is achieved by using three more meta cells[24], in which the PCE was low in blue or red regime due to the partial overlap of the low efficiency bands. In our approach, phase control in metasurfaces was achieved through the superposition of cubic phase and Fresnel phase. The single structure size and shape was designed to achieve a theoretical RGB PCEs

exceeding 76% and the actual PCEs measured beyond 60%. The propagation properties of the CAFAB metasurface, as depicted in Fig. 3, validate its effectiveness for operation at RGB color regime. Discrepancies between simulation and experimental results might be attributed to the nanofabrication related resolution limitations and variations in the experimental optical path. With such a high PCE, the metasurface developed in this work can also find important applications in integrated photonic chips[20–22], holographic color imaging[16,17], etc.

Albeit a challenging underwater environment, the developed metasurface-based multi-wavelength circular auto-focusing Airy beam transmitter exhibits exceptional potential for constructing robust, high-capacity, and long-range UWOC links in various application scenarios. Theoretical modeling (see Figs. S2 and S3 in the Supporting Information) demonstrates the self-healing effect and improved stability in transmittance of the CAFAB beam when obstructed at different positions, surpassing the performance of the Gaussian beams. The resistance of Airy beams against obstacles and bubble disturbances were further assessed by a detailed experimental analysis of the UWOC system performance. The distinctive intensity distribution of CAFAB and its self-healing properties result in superior anti-disturbance stability compared to Gaussian beams (Figs. 4 and 5). Moreover, the efficient metasurface minimizes optical losses at the transmission end, and the utilization of multiple wavelengths allows for adaptation to varying water quality environments and transmission distance (Figs. S1 and 5). The total data rate achieved through multi-wavelength transmission exceeds 20 Gbps over an equivalent distance of 100 m, highlighting the substantial transmission capacity of our adaptive system. Then, a 4 K ultra-high-resolution video robust transmission is demonstrated in Fig. 6. CAFAB based UWOC link enable the error-free 4 K video streaming reception in the challenging channel, because CAFAB based system exhibits slight BER fluctuations, satisfying the threshold for FEC decoding (Fig. S7). On the other hand, the results of the received beam quality reveals that CAFABs have a more concentrated centroid distribution with higher optical power than Gaussian beams (Fig. S8). Further utilizing the optical adaptive

compensation[28–30] for beam recovery, this scheme is expected to become one of the solutions for underwater wireless communication links in the 6G era.

In conclusion, this work presents a method for designing and fabricating a high-efficiency visible ultra-broadband CAFAB metasurface. Meanwhile, the work significantly expands the research scope of UWOC systems by achieving stable, >20 Gbps data transmission in complex channels. In the future, integration of laser modules with metasurfaces will further reduce system power loss. By combining injection locking[40] and WDM techniques, the capacity and range of UWOC systems can be exponentially enhanced, paving the way for practical applications of UWOC. Our work facilitates the development of visible ultra-broadband circular self-focusing Airy beam metasurface, and provides a solution for reliable and high-speed underwater wireless data links.

## Methods

### Numerical simulations

The meta is numerically calculated using COMSOL Multiphysics® (ref. 41) based on the time-domain element method. In the simulations, periodic boundary conditions are used for the calculations on the sidewalls of the metasurfaces. Perfectly matched layers (PML) are added where the beam enters and exits. The refractive indices of $SiO_2$ and $TiO_2$ are measured by ellipsometry.

In the calculation of the light field, the Rayleigh Sommerfeld diffraction integral is used for numerical calculation:

$$U_1(\mathbf{r}_1) = \frac{1}{i\lambda} \int_{S_A} U_0(\mathbf{r}_0) T(\mathbf{r}_0) \frac{\exp(ik|\mathbf{r}_{01}|)}{|\mathbf{r}_{01}|} \cos(\mathbf{r}_{01}, \mathbf{n}) dS \quad (2)$$

Where $U_1$ is the receiving surface, $U_0$ is the transmitting surface, $T$ is the transmittance of the transmitting surface, the coordinates of the transmitting surface and the receiving surface are $\mathbf{r}_0$ and $\mathbf{r}_1$ respectively, the vector $\mathbf{r}_{01}$ is the distance between two points, and the vector $\mathbf{n}$ is the unit distance. Numerical calculations can be performed by integrating the surface $S_A$.

### Fabrication of the metasurface

Firstly, the 13 nm indium tin oxide (ITO) on glass substrate is ultrasonically cleaned. The 800 nm TiO2 film is deposited on the ITO coated glass substrate by using an electron beam (Ebeam) evaporator (SKE_A_75). The refractive index, extinction coefficient, and thickness are measured by ellipsometer. The 100 nm electron-beam resist (PMMA) is spin-coated at 4000 rpm on the film and baked at 180 °C for 1 h. The resist is patterned by electron-beam lithography (Raith E-line) at a dose of 90 μC/cm² at an accelerating voltage of 30 keV. After immersing in MIBK&IPA (1:3) solution for 30 s, a reverse pattern is generated on the surface. A layer of 23 nm chromium (Cr) is deposited on the sample using Ebeam evaporation. After the samples is soaked in remover PG solution (Micro Chem) for 12 h, the pattern was then transferred to the TiO2 film by reactive ion etching (Oxford Plasmalab System 100). Finally, the Cr hard mask was removed by soaking in a Cr etchant solution (Aldrich Chemistry) for 10 min.

### UWOC system based on RGB Airy beams

In the experiment, the signal is modulated as discrete multi-tone (DMT) format with bit power allocation according to channel characteristics, which can approach the Shannon limit of the system. And its BER just satisfies FEC limit. Figure S4 in the Supporting Information shows the detail of the UWOC system setup. In the electrical domain, transmitting DSP imports modulated digital signal into the arbitrary waveform generator (AWG) for high-speed DA conversion, and the sampling frequency is set to 8 Gsa/s. Then electrically amplified RF signal is directly modulated to the tri-color lasers through a bias Tee. In the processing of the optical domain, the collimated linearly polarized Gaussian beam is converted into a left-handed circular polarization by achromatic quarter-wave plate (QWP). The designed metasurface supports more than 60% conversion efficiency for both three RGB wavelengths. In order to obtain pure right-handed circular Airy beam after conversion, a combination of QWP and liner polarizer (LP) is used to filter out the unconverted Gaussian beam. The underwater channel is a 2 m water-tank. And opaque black paint is used to simulate underwater obstacles of different sizes, which are normalized to -14 mm ($\omega_0$) transmitter beam apertures. A speed-adjustable air pump (10 L/min maximum) is fixed at the 1/4 position of the water tank to generate different bubble conditions.

## Data availability

The authors declare that the main data supporting the findings of this study are available within this paper and its Supplementary Information files. The source data behind Figs. 2c, 3a(spectra), b–d, 4a–d, 5, 6b and Supplementary Figs. S1b, c, S3b, c, S5–7, S8a, b are provided in the Source Data files, and extra data are available from the corresponding author on request. Source data are provided with this paper.

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

## Acknowledgements

This research is partially funded by National Key Research and Development Program of China (2022YFB2802803), Natural Science Foundation of China Project, grant numbers 61925104, 62031011, 62274042; Natural Science Foundation of Shanghai, grant number 21ZR1406200; The joint project of China Mobile Research Institute & X-NET; The Key Research and Development Program of Jiangsu Province (BE2021008-5).

## Author contributions

N.C., S.X. and Q.S. give the research direction, and contributed the basic framework and feasible technical route of the project. J.H., Z.G. and C.S. design specific research content according to the idea. C.S. supervised the implementation progress of the research. Z.G. designed and proposed the multi-wavelength visible circular auto-focusing Airy beam metasurface with the assistance of X.J. and Q.C. S.Y., Z.H. and H.C. provided the standard experimental site and equipment for underwater optical wireless communication. J.H. completed the construction and testing of the entire experimental system. J.S. analyzed and guided the adjustment of experiments. J.H. and Z.G. co-wrote the manuscript and revised it based on comments from all authors.

## Competing interests

The authors declare no competing interests.
