## [Peer Review File · Nature Communications]

A metasurface-based full-color circular auto-focusing Airy beam transmitter for stable high-speed underwater wireless optical communicationsREVIEWER COMMENTS

Reviewer #1 (Remarks to the Author):

In the manuscript entitled "A metasurface-based full-color circular auto-focusing Airy beam transmitter for stable high-speed underwater wireless optical communications", the authors designed a full-color circular auto-focusing Airy beams metasurface transmitter to realize reliable, large-capacity, and long-distance underwater wireless optical communication links.

Underwater wireless communication is an important research field and attracts many research interests. The authors claimed that they proposed an adaptive underwater wireless optical communication link and experimentally evaluated the performance. However, both the design principles of the ultra-broadband Airy beam metasurface (Refs. 23 and 24) and the performance evaluation methods have been reported before (Ref. 35). Therefore, I think the novelty of the presented concept is not enough to publish in Nature Communications. I suggest the authors conduct some experiments to directly show the communication ability of the proposed system.

There are some minor issues to help improve the manuscript:

- (1)The authors used quite a lot of abbreviations throughout the manuscript, some of them are unnecessary (IoUT, FWHM, SLM, SNR). Too many abbreviations may confuse the readers, especially non-expert readers.
- (2)Figure 1 is very complex and there are few illustrations about Fig. 1 in the main text.
- (3)As claimed by the authors, the experimental results agree with the simulation results in Fig. 3. However, the experimental results of RGB Airy field distribution at the focus position (Fig. 3d) are quite different from the simulated results (Fig. 3c).

Reviewer #2 (Remarks to the Author):

Dear Collgues,

I read your valuable work and I am convinced that it deserves to be published. In fact, it is original and clear with all single sections detailed explained and excellent results that confirm the theoretical approach. Nevertheless in order to better describe the final version and therefore the readability of the paper I would like to suggest some possible improvements.

- 1) First: it is very original the presentation of the order of the paragraphs; I mean, typically, the Results are presented after "materials and methods", but this way to present is very effective therefore, if the Editor is agree, my opinion is to leave the paper in this form. Moreover, in the paper lacks a Conclusion paragraph, which is typically very useful for understanding the value of a paper.
- 2) Joined with previous point, the Abstract should be written differently; I mean: in the abstract should be introduced the work anticipating the value of the work without generic comments that are typically reserved to the Conclusion. So e.g., this sentence: "Dear Collgues,

I read your valuable work and I am convinced that it deserves to be published. In fact, it is original and clear with all single sections detailed explained and excellent results that confirm the theoretical approach. Nevertheless in order to better describe the final version and therefore the readability of

the paper I would like to suggest some possible improvements.

1) First: it is very original the presentation of the order of the paragraphs; I mean, typically, the Results are presented after "materials and methods", but this way to present is very effective therefore, if the Editor is agree, my opinion is to leave the paper in this form. Moreover, in the paper lacks a Conclusion paragraph, which is typically very useful for understanding the value of a paper.

2) Joined with previous point, the Abstract should be written differently; I mean: in the abstract should be introduced the work anticipating the value of the work without generic comments that are typically reserved to the Conclusion. So e.g., this sentence: "our work facilitates the development of visible ultra broadband circular self-focusing Airy beam metasurface, and provides a solution for reliable and high-speed underwater wireless data links.", in my opinion, should be inserted in the Conclusion. It is possible to substitute it with a more general sentence.

3) Considering the value of the paper, if the length allow it, in my opinion a paragraph that well describe the measurement bench should be introduced in the paper, transferring la figura S4 and relative comments in the paper and not only in the support materials.

4) Some figures should improve their resolutions, in particular Figures 3, 4 and 5.

5) I suggest to read this paper that resume many UWOC strategies: Schirripa Spagnolo, G., Cozzella, L., Leccese, F. Underwater confirmter Optical Wireless Communications: Overview (2020) Sensors (Basel, Switzerland), 20 (8), . DOI: 10.3390/s20082261. WOS:000533346400103, eid=2-s2.0-85084784594.

Response Letter

Nature Communications (Manuscript ID: NCOMMS-23-42666)

A metasurface-based full-color circular auto-focusing Airy beam transmitter for stable high-speed underwater wireless optical communications

We would like to express our sincere gratitude to all reviewers for the valuable feedback and suggestions on our work. The insightful comments prompted us to make further improvements to the manuscript.

In this letter, we provide a detailed point-by-point response to each comment from the reviewers. To facilitate the distinction of figures in different files, we have marked them as follows: Fig. R1-4 refer to the figures in the response letter; Fig. 1-6 are the figures in the manuscript; Fig. S1-8 are the figures in supporting information; [1] refers to the references cited in the response letter; Ref. 23 refers to the paper cited in the manuscript. A revised manuscript with changes made in red color have also been provided for clarity.

Reviewer #1

General Comment: *In the manuscript entitled “A metasurface-based full-color circular auto-focusing Airy beam transmitter for stable high-speed underwater wireless optical communications”, the authors designed a full-color circular auto-focusing Airy beams metasurface transmitter to realize reliable, large-capacity, and long-distance underwater wireless optical communication links.*

Underwater wireless communication is an important research field and attracts many research interests. The authors claimed that they proposed an adaptive underwater wireless optical communication link and experimentally evaluated the performance.

Response to General Comment: We would like to thank your guidance and advices for our work. We have carefully considered each comment and provided a detailed point-by-point response as follows:

Comment 1: *Both the design principles of the ultra-broadband Airy beam metasurface (Refs. 23 and 24) and the performance evaluation methods have been reported before (Ref. 35). Therefore, the novelty of the presented concept is not enough to publish in Nature Communications.*

Response to Comment 1: Thank you for your comment. We would like to provide a more detailed discussion regarding the significance of our work in relation to Refs. 23, 24 and 35. It appears that the emphasis on the significance of our work and the innovative applications in the demonstrated system may not have been adequately highlighted. We would like to address your concerns related to the novelty of our work in the following discussion.

In Ref. 23, a two-dimensional square Airy beam metasurface using phase modulation within the wavelength range of 550 nm to 700 nm has been realized. Noting that they use the cubic phase and Fresnel phase to synthesis the phase distribution of metasurface, as shown in function (1) [1]-[3]. In addition, Airy beam metasurfaces can also be realized by the 3/2-phase method [4] or amplitude modulation [5].

$$\Phi(x_{meta}, y_{meta}) = \frac{(2\pi b_1)^3}{3} (x_{meta}^3 + y_{meta}^3) - \frac{\pi}{\lambda f} (x_{meta}^2 + y_{meta}^2) \quad (1)$$

Similar to our design approach, the advantage of synthetic-phase metasurface is the flexible utilization of the focal length of a Fresnel lens in the design. This allows for the generation of a series of Airy beams with controllable focal lengths, narrow beam widths, and extended propagation distances. Importantly, these beam parameters are fully adjustable, making these tunable focal length Airy beams particularly attractive for applications in high-resolution, wide-field imaging, and deep-penetration optical operations. However, this paper does not showcase the application of these designs in practical scenarios but rather provides a prospect for future work. In our study, although we employ the synthetic-phase method, our objective is explicitly defined: the application in underwater optical communication systems using metasurface transmitter. Moreover, by further refining the use of polar coordinates for synthetic-phase to generate circular self-focusing Airy beams, the main lobe radius and focusing distance are explicitly reflected in the phase function (2).

$$\varphi_{meta}(r) = (2\pi b_1)^2 (r - r_0)^3 / 3 - 2\pi / \lambda \left[\sqrt{(r - r_0)^2 + f^2} - f \right] \quad (2)$$

By setting the focal position f to 1 mm and the main lobe radius r_0 to 55 μm , the conversion from Gaussian light to Airy beams can be achieved within a distance of less than 5 mm. This value is remarkable as our approach eliminates the need for additional compensation for the working distance. Additionally, circular, small footprint, and high-resolution focused Airy beams offer advantages in underwater communication, sensing and imaging. This is particularly beneficial as it reduces the cost of large-aperture transmitting and receiving lenses.

In terms of the design method for circular Airy beams in Ref. 24, they use four meta-cells on one sample to respond to different wavelength bands in the visible spectrum. The metasurface phase distribution is obtained using the 3/2-phase modulation method, expressed in function (3) [6].

$$\varphi(x, \lambda) = -\frac{4}{3} k \sqrt{a} (-x)^{3/2} + \varphi_{shift}(\lambda) \quad (3)$$

This work also does not provide a comprehensive conceptual framework from design to application domains. Meanwhile, results show the four cells based metasurfaces exhibit lower conversion efficiency at the edges of the response spectrum. Particularly, there is a significant difference in the response of these meta-cells around 600 nm, and these differences tend to further widen with increasing wavelength, which may result in the complete inoperability of one or two cells. In our work, due to the design and optimization of the structural unit, one single meta-cell can be used to achieving an ultra-broadband response. This approach not only reduces the complexity of the fabrication process but also improve the efficiency of meta-cell at edge of the band. From an application perspective, we firstly applied this circular auto-focusing Airy beam to resist random disturbances in UWOC links, which further enhances the practical value of structured light.

Fig. R1. Polarization conversion efficiency versus wavelength or frequency. (a) Ref. 23 [1], (b) Ref. 24 [6], The colors of the line corresponding to the dimensions of the unit-cell are red (90, 230 nm), blue (100, 255 nm), black (110, 280 nm), green (125, 310 nm). (c) our work.

Fig. R1 shows the polarization conversion efficiency (PCE) demonstrated in Refs. 23, 24 and our work. The comparative results clearly demonstrate that our work exhibits a broader operating wavelength range (450-640 nm) while maintaining a more stable PCE in working band. For the former, a broader spectrum allows us to simultaneously cover the emission wavelengths corresponding to red (637 nm), green (517 nm), and blue (455 nm) (RGB) lasers, making it suitable for our tri-color integrated laser module, which is an aspect that the other two works cannot achieve. For the latter, the uniformity of PCE is of significant importance in the detection in optical communication systems. For instance, Si-based photodetectors (PDs) often exhibit non-uniform spectral

responses, requiring additional optical color filters to compensate for this non-uniform response to achieve consistent detection sensitivity [7]. However, this comes at the cost of power loss. In our work, the metasurface exhibits uniformly high PCEs in the visible color regime. Such device enables full-color operation, which is of unique advantages for optical communications by eliminating the need for additional power allocation schemes.

In Ref. 35, they studied the performance of Bessel beam and orbital angular momentum (OAM) beam generated using a bulky spatial light modulator (SLM). Such technology to generate structured light results in limited beam resolution and quality. And the experimental results do not indicate a significant improvement of system performance under challenging channel conditions by using Bessel beam or OAM beam. In our work, we posit that employing metasurface based Airy beam holds greater potential for enhanced transmission when strong disturbance is presented in the channel. The intensity distribution of Airy beam exhibits resilience to perturbations based on numerical analysis [13]. Also, metasurface can generate Airy beams with high resolution, making our technology suitable for photonics integration. Therefore, we believe our work represents the first experimental validation of the significant potential of circular auto-focusing Airy beams in challenging UWOC environments.

Indeed, since the emergence of Airy beams, their unique attributes such as self-healing, non-diffracting, and self-accelerating properties have demonstrated significant potential in particle manipulation [8], high-resolution microscopy [9][10], and optical bullets [11]. The highlights of these works include their ability to design a unique structure towards application scenarios and substantiating their claims through comprehensive experimental validations. We believe our work shares similar quality, as it essentially involves the design of a full color metasurface-based circular auto-focusing Airy laser transmitter and demonstration of underwater wireless optical communication (UWOC) applications in challenging environments. The findings are significant to address one of the major challenges for practical UWOC deployments, with the full color transmitter design as a noteworthy aspect of our work.

Simultaneously, we have expanded the application dimensions of Airy beams.

Overall, we believe our work possesses impactful innovations which justifying publication in *Nature Communications*:

1. Design a circular self-focusing Airy beam metasurface covering the entire visible spectrum (RGB).
2. Applicable for integrated tri-color laser modules, with high and closely-matched PCEs across the visible spectra (455 nm: 79.6%, 517 nm: 90.2%, 637 nm: 76.0%).
3. Expand the application scope of Airy beams, providing the first experimental validation in UWOC systems and demonstration of 4K video streaming applications.
4. Present a unique solution for achieving a stable, high-capacity, long-distance UWOC link.

Revision to Comment 1: We make the following modifications to the Introduction section of the main text based on the feedback:

“... Recently, generation of visible Airy beams by using metasurface has been studied towards broad-band, full-color and high-efficiency. Employing synthetic-phase method contributes to producing tunable focal length red Airy beam²³. Fabricating various meta-cells can be utilized to collectively cover the 450-600 nm response band²⁴. However, it is still challenging to achieve a high conversion efficiency across the visible color regime using a single structure. In optical communication and imaging systems, a uniform spectral response in optical devices is imperative, as it eliminates the need for additional power distribution or filters....”

“... Simultaneously, OAM or Bessel beams based on spatial light modulators have not demonstrated significant advantages in mitigating the impact of underwater bubbles³⁶....”

Comment 2: *The authors should conduct some experiments to directly show the*

communication ability of the proposed system.

Response to Comment 2: Thank you for your advice. Considering the practical application performance of our proposed technology, we have added the following more intuitive communication performance demonstration experiments based on your suggestions.

Video transmission or display is crucial in underwater wireless networks, especially for applications like real-time imaging in underwater exploration or 3D reconstruction of underwater scenes. With the development of 6G networks, there is an increased demand for enhanced transmission metrics, including ultra-large capacity (4K ultra-high-resolution video streaming), ultra-low system latency, and exceptional system robustness. Here, we present a 4K ultra-high-resolution video transmission based on our high-speed full-color metasurface laser transmitter in UWOC data link. The experimental results clearly present the enhancement of robustness against underwater bubbles and obstacles by using our full-color metasurface transmitter. The basic framework is illustrated in Fig. R2(a). The 4K video source is transformed into a transmittable bitstream signal through frame encoding. WDM technology enables high-speed transmission of the bitstream through RGB channels. The forward error correction (FEC) coding is adopted to enhance the error-correcting capabilities of the video bitstream. After FEC encoding, the data is mapped to the 16-QAM modulation format for channel transmission. Beam shaping into RGB CAFAB implemented by full-color metasurface significantly enhances the stability of the communication link, especially in the presence of underwater obstacles and bubbles (the water environment adopts water II, i.e., $0.43 \omega_0$ obstacle with middle bubbles). This allows the receiver to decode the 4K video data.

Fig. R2(b) shows a comparison of the quality metrics for 4K video transmitted by CAFAB (with metasurface) and Gaussian beam (without metasurface) based systems. Here, we used two video quality metrics to evaluate the transmission results of 4K video: peak signal-to-noise ratio (PSNR) and structural similarity (SSIM). PSNR is a metric

used to measure the quality of an image or video, commonly employed to assess the difference between a distorted image or video and its original version.

Fig. R2. Underwater 4K video streaming using the full-color metasurface. **a** Schematic of the system. **b** The quality metrics, PSNR and SSIM of each frame received by the CAFAB (with metasurface) or Gaussian beam (without metasurface) based UWOC system versus time. **c** The received video frames with and without the full-color metasurface at different frame sequences.

The formula for PSNR are as follows:

$$PSNR = 10 \cdot \log_{10} \left(\frac{MAX^2}{MSE} \right) \quad (4)$$

MAX represents the maximum possible pixel value in the image or video (typically 255 for 8-bit images). MSE stands for Mean Squared Error, calculated as the average of the squared differences between corresponding pixels. A higher PSNR value indicates a smaller difference between the distorted and original images or videos, signifying better quality.

SSIM serves as a comprehensive, full-reference image quality assessment metric, evaluating image similarity across luminance l , contrast c , and structure s , which expressed by (5) and (6).

$$l(X, Y) = \frac{2\mu_X\mu_Y + C_1}{\mu_X^2 + \mu_Y^2 + C_1}, \quad c(X, Y) = \frac{2\sigma_X\sigma_Y + C_2}{\sigma_X^2 + \sigma_Y^2 + C_2}, \quad s(X, Y) = \frac{\sigma_{XY} + C_3}{\sigma_X\sigma_Y + C_3} \quad (5)$$

$$SSIM(X, Y) = l(X, Y) \cdot c(X, Y) \cdot s(X, Y) \quad (6)$$

Where μ_X and μ_Y represent the means of images X and Y , σ_X and σ_Y denote the variances of images X and Y , and σ_{XY} represents the covariance between images X and Y . The SSIM index ranges from 0 to 1, where a higher value indicates less distortion in the images.

Results of Fig. R2(b) significantly demonstrate the superior robustness of the CAFAB based system. The PSNR of each frame of the 4K video remains around 43 dB, and the SSIM is consistently above 99.3%. The Gaussian beam based UWOC link, due to greater attenuation from obstacles and larger power fluctuations caused by bubbles, can only recover parts of the frame images, as shown in Fig. R2(c). Its average PSNR and SSIM are 19.62 dB and 49.02% respectively.

Next, we will present the other technical details involved in the experiment and intend to include these results in the Supplementary Information materials.

Fig. R3. BER fluctuates over time in UWOC systems based on CAFAB or Gaussian beams: (a) pre-FEC, and (b) post-FEC.

We conducted a BER analysis of the received video streaming, including pre-FEC and post-FEC BER, as displayed in Fig. R3. Fig. R3(a) shows the statistical results of pre-FEC BER. The BERs of the Gaussian beam-based system fluctuate dramatically over time, ranging from 5.7×10^{-3} to 2.9×10^{-2} with an average of 1.4×10^{-2} . The pre-FEC BER of the CAFAB based system is more stable, averaging at 3.7×10^{-3} . These results are in

consistent with the statistical outcomes for data rate and ROP presented in Fig. 4 and Fig. 5 in the manuscript. When using the 10% low-density parity-check (LDPC) code as FEC encoding, all errors generated by the CAFAB based system can be corrected, resulting in no errors at the receiver, shown in Fig. R3(b). However, the Gaussian beam-based system can only recover the signal without errors at certain moments, which is already reflected in Fig. R2(c).

Fig. R4. The beam quality of the received CAFAB and Gaussian beam in complex underwater channel. Scatter plot of centroid position changes in beam clusters of (a) CAFAB, (b) Gaussian beams. c Selected CCD images of received CAFAB and Gaussian beams.

In addition, we monitored and compared the beam quality differences of received RGB CAFAB and Gaussian beam in 4K video communication system, as shown in Fig. R4. In terms of the variations in the centroid positions of the received beam clusters, the CAFAB clusters exhibit a more stable intensity distribution, with centroid position drift ranging only between -0.05 and 0.05 on the normalized x-y axes and an average offset of 1.6×10^{-2} , as shown in Fig. R4(a). For Gaussian beam clusters, the range of centroid

movement extends to between -0.1 and 0.1 on the biaxial scale, with twice the offset of CAFAB, 3.0×10^{-2} , as shown in Fig. R4(b). These results also directly demonstrate the superior robustness of the CAFAB based UWOC system when facing underwater bubble interference. Selected received RGB beam CCD images are displayed in Fig. R4(c), showing that CAFAB reshaped by the full-color metasurface maintains higher and more stable ROP when faced with obstructions and bubbles. The average ROP of the RGB channels in the CAFAB based system, compared to the Gaussian beam-based system, has increased by 2.57dB, 3.48 dB, and 7.31dB respectively. This provides significant assurance for 20 Gbps 4K video transmission in the complex underwater environment.

Overall, we compared the performance of traditional Gaussian links with our proposed full-color metasurface CAFAB links in complex underwater environments for 4K video streaming. The results demonstrate the superior robustness of the CAFAB system, capable of error-free recovery of each frame at the receiver. These findings vividly illustrate the broad application prospects of the full-color metasurface transmitter in future 6G wireless optical networks.

Revision to Comment 2: We incorporate the content of Fig. R2 into Fig. 6 of the main text and the relevant description, while the contents of Fig. R3 and Fig. R4 are added in Supplementary Information (Fig. S7 and Fig. S8).

“Fig. 6 present a 4K ultra-high-resolution video transmission based on our high-speed full-color metasurface laser transmitter in the UWOC data link. The experimental results clearly present the enhancement of robustness against underwater bubbles and obstacles by using the metasurface. The system structure is illustrated in Fig. 6(a). The 4K video source is transformed into a transmittable bitstream signal through frame encoding. WDM technology enables high-speed transmission of the bitstream through RGB channels. Beam shaping into RGB CAFAB implemented by full-color metasurface significantly enhances the stability of the communication link in water II. This allows the receiver to decode the 4K video data. Fig. 6(b) shows a comparison of the quality

metrics for 4K video received by CAFAB (with metasurface) and Gaussian beam (without metasurface) based systems. Results significantly demonstrate the superior robustness of the CAFAB based system. The peak signal-to-noise ratio (PSNR) of each frame of the 4K video remains around 43 dB, and the structural similarity (SSIM) is consistently above 99.3%. While in the Gaussian beam based UWOC link, the system can only recover parts of the frame images, as shown in Fig. 6(c). The corresponding average PSNR and SSIM are 19.62 dB and 49.02%, respectively. The results suggest that significant attenuation from obstacles and larger power fluctuations caused by bubbles was presented in the system without metasurface. The advances of the CAFAB based system pave the path for robust, high-capacity UWOC links.”

“.... Then, a 4K ultra-high-resolution video robust transmission is demonstrated in Fig. 6. CAFAB based UWOC link enable the error-free 4K video streaming reception in the challenging channel, because CAFAB based system exhibits slight BER fluctuations, satisfying the threshold for FEC decoding (Fig. S7). On the other hand, the results of the received beam quality reveals that CAFABs have a more concentrated centroid distribution with higher optical power than Gaussian beams (Fig. S8). Further utilizing the optical adaptive compensation²⁸⁻³⁰ for beam recovery, this scheme is expected to become one of the solutions for underwater wireless communication links in the 6G era.”

Comment 3: *The authors used quite a lot of abbreviations throughout the manuscript, some of them are unnecessary (IoUT, FWHM, SLM, SNR). Too many abbreviations may confuse the readers, especially non-expert readers.*

Response to Comment 3: Thank you for your valuable comment. We agree that reducing the use of abbreviations for these technical terms would enhance the readability of this paper and make it more accessible to a broader audience. We have replaced the abbreviations of these technical terms with their corresponding full names, such as IoUT - Internet of Underwater Things, FWHM - full widths at half maximum, SLM - spatial light modulator, SNR - signal-to-noise ratio.

Comment 4: *Figure 1 is very complex and there are few illustrations about Fig. 1 in the main text.*

Response to Comment 4: Thank you for your advice. Fig. 1a depicts the emerging underwater wireless optical communication link based on the full-color Airy laser metasurface emitter proposed in this work. As a comparison, conventional UWOC scheme is presented in Fig. 1b. The intensity distribution of the two beams is shown in Fig. 1(i) and Fig. 1(ii). After transmission in underwater channel, the characterization of the received signal by the photodetector (PD) is displayed in Fig. 1(iii) and Fig. 1(iv). We have added explanatory details regarding Fig. 1.

Revision to Comment 4:

“.... Fig. 1 presents the framework and performance comparison between traditional UWOC link and our proposed adaptive link (Fig. 1a). In traditional links (Fig. 1b), blue (~450 nm) band is considered to be the best communication wavelength for UWOC due to its smallest attenuation coefficient of 0.007 m^{-1} under pure water.... Fig. 1(ii) shows the Gaussian beams intensity distribution. When the obstacles present in the underwater channel, there will be a significant optical loss, leading to a massively reduced ROP. Also, air-water turbulence due to the presence of bubbles in UWOC link causes severe disturbances in the ROP, as shown in Fig. 1(iv)....”

“In order to support high-reliability, large-capacity and long-distance UWOC system, an adaptive scheme using tri-color Airy beams is proposed (Fig. 1a).... RGB CAFAB are generated by employing our designed ultra-broadband metasurface, the intensity distribution of the beam is illustrated in Fig. 1(i).... In the region of self-focusing, the obstruction resulting from small-scale obstacles is mitigated by the energy carried in the side lobes. Within the divergent region, the system's adaptability to obstacles is improved due to the distinctive circular energy distribution of Airy beam. In the presence of air bubbles, the intensity distribution of the Airy beam exhibits robustness against perturbations³⁹. The received beam and data signals are shown in Fig. 1(iii).”

Comment 5: *As claimed by the authors, the experimental results agree with the*

simulation results in Fig. 3. However, the experimental results of RGB Airy field distribution at the focus position (Fig. 3d) are quite different from the simulated results (Fig. 3c).

Response to Comment 5: Thank you for your question. We agree that the statement in the current manuscript may confuse the reader and have revised accordingly. In fact, Fig. 3d depicts the light field distribution when the Airy beam is initially generated at $z = f$, corresponding to the position of the white dash in Fig. 3a. The corresponding simulation content is shown in Fig. 3b. Fig. 3c is the simulation result at the focal position of the Airy beam, which correspond to the position of the yellow dash in Fig. 3a.

In our current experimental setup, a pinhole is used to improve the beam quality of the edge-emitting laser diode. However, such pinhole cannot be too small as the light output power would be limited. The tuning of the pinhole size may lead to the differences in the side lobes of the Airy beam presented in Fig. 3d and Fig. 3b. Additionally, there might be some alignment deviations in the optical path, causing certain intensity non-uniformity on the ring of Airy beam pattern. Nonetheless, the measured position and the full width at half maximum of the main lobe are well aligned with the simulation results, indicating that the metasurface functions as expected.

Revision to Comment 5: We have made corresponding modifications to the caption and description of Fig. 3:

“Fig. 1 CAFAB simulation and measurement for the wavelengths of 455 nm, 517 nm and 637 nm. a Left part is the x-z section of the Airy beam propagation trajectory simulation with the $z = 0$ plane as the metasurface plane, right part are the spectra of RGB lasers in log scale. b-c Simulation of x-y RGB Airy field distribution (b) at position $z = f$ (white dash), (c) at the focus position $z = 4.6$ mm, 4.7 mm, 4.8 mm, respectively (yellow dash). d Experimental measurement of RGB Airy field distribution at the position $z = f$.”

“To verify the effectiveness of the design process, we prepare and numerically demonstrate a metasurface that can generate RGB circular self-focusing Airy beams. Considering the computational constraints of the simulation work, the r_0 , b_1 and focal

length f are set to $55 \mu\text{m}$, $0.01 \mu\text{m}^{-1}$ and 1mm , respectively....”

“CAFAB metasurface was fabricated according to the phase profile design of the synthetic metasurface. The phase profile consists of the cubic phase ($b_1 = 0.01 \mu\text{m}^{-1}$), the Fresnel holographic lens phase (focal length $f = 1 \text{mm}$)....”

Reviewer #2

General Comment: *I read your valuable work and I am convinced that it deserves to be published. In fact, it is original and clear with all single sections detailed explained and excellent results that confirm the theoretical approach. Nevertheless, in order to better describe the final version and therefore the readability of the paper I would like to suggest some possible improvements.*

Response to General Comment: We deeply appreciate your recognition of our work and are grateful for your valuable suggestions. We believe that the quality and readability of our article can be enhanced based on your feedback. Next, we will respond to your comments point by point:

Comment 1: *First: it is very original the presentation of the order of the paragraphs; I mean, typically, the Results are presented after "materials and methods", but this way to present is very effective therefore, if the Editor agree, my opinion is to leave the paper in this form. Moreover, in the paper lacks a Conclusion paragraph, which is typically very useful for understanding the value of a paper.*

Response to Comment 1: Thank you for your comment. We fully understand your need for a particular order of paragraphs in the manuscript, but unfortunately, submissions to the Nature series of journals must adhere to the manuscript formatting requirements outlined in their standard documents. Additionally, concerning the concluding paragraph of the article, we previously placed it at the end of the Discussion section, which is typically the standard practice for *Nature Communications*.

The concluding paragraph is: *"In conclusion, this work presents a method for designing and fabricating a high-efficiency visible ultra-broadband CAFAB metasurface. Meanwhile, the work significantly expands the research scope of UWOC systems by achieving stable, >20 Gbps data transmission in complex channels. In the future, integration of laser modules with metasurfaces will further reduce system power loss. By combining injection locking⁴⁰ and WDM techniques, the capacity and range of*

UWOC systems can be exponentially enhanced, paving the way for practical applications of UWOC. Our work facilitates the development of visible ultra-broadband circular self-focusing Airy beam metasurface, and provides a solution for reliable and high-speed underwater wireless data links.”

Comment 2: *Joined with previous point, the Abstract should be written differently; I mean: in the abstract should be introduced the work anticipating the value of the work without generic comments that are typically reserved to the Conclusion. So e.g., this sentence: "our work facilitates the development of visible ultra-broadband circular self-focusing Airy beam metasurface, and provides a solution for reliable and high-speed underwater wireless data links.", in my opinion, should be inserted in the Conclusion. It is possible to substitute it with a more general sentence.*

Response to Comment 2: Thank you for your comment. We agree with your perspective. Regarding the expression in the abstract, we will revise it to:

“Due to its unique intensity distribution, self-acceleration, and beam self-healing properties, Airy beam holds great potential for optical wireless communications in challenging channels, such as underwater environments. As a vital part of 6G wireless network, the Internet of Underwater Things requires high-stability, low-latency, and high-capacity underwater wireless optical communication (UWOC). Currently, the primary challenge of UWOC lies in the prevalent time-varying and complex channel characteristics. Conventional blue Gaussian beam-based systems face difficulties in underwater randomly perturbed links. In this work, we report a full-color circular auto-focusing Airy beams metasurface transmitter for reliable, large-capacity and long-distance UWOC links. The metasurface is designed to exhibit high polarization conversion efficiency over a wide band (440-640 nm), enabling an increased data transmission rate of 91% and reliable 4K video transmission in wavelength division multiplexing (WDM) based UWOC data link. The successful application of this metasurface in challenging UWOC links establishes a foundation for underwater interconnection scenarios in 6G communication.”

Comment 3: *Considering the value of the paper, if the length allows it, in my opinion a paragraph that well describe the measurement bench should be introduced in the paper; transferring la figure S4 and relative comments in the paper and not only in the support materials.*

Response to Comment 3: Thank you for your suggestion. Due to length limit of Nature Communications, it is difficult to include details about the measurement bench in the manuscript. According to the authors' guidance, we would prefer to include those details and the flowchart of the experimental platform in supplementary information, such as in Fig. S4.

Comment 4: *Some figures should improve their resolutions, in particular Figures 3, 4 and 5.*

Response to Comment 4: Thank you for your advice. We have provided high resolution figures. Figures with a resolution of 300 dpi have also been uploaded separately to the editorial team.

Comment 5: *I suggest to read this paper that resumed many UWOC strategies: Schirripa Spagnolo, G., Cozzella, L., Leccese, F. Underwa to confirmter Optical Wireless Communications: Overview (2020) Sensors (Basel, Switzerland), 20(8), DOI: 10.3390/s20082261. WOS:000533346400103, eid=2-s2.0-85084784594.*

Response to Comment 5: Thank you for providing us with this reference, which provides a good overview of UWOC technology. The paper discussed the application, optical properties of water environments, the basic framework of UWOC systems, and also forecasts future technological trends. We have included this paper in our manuscript:

35. Schirripa, S. G., Cozzella, L. & Leccese, F. Underwater optical wireless communications: Overview. Sensors. 20, 2261 (2020).

Reference

[1]. Wen, J. et al. All-dielectric synthetic-phase metasurfaces generating practical airy beams. *ACS nano*.

-
- 15**, 1030-1038 (2021).
- [2]. Ling, J.; Yang, Q.; Zhang, S.; Lu, Q.; Liu, S.; Guo, C. Improved Generation Method Utilizing a Modified Fourier Spectrum for Airy Beams with the Phase-Only Filter Technique. *Appl. Opt.* 2017, **56**, 7059.
- [3]. Latychevskaia, T.; Schachtler, D.; Fink, H.-W. Creating Airy Beams Employing a Transmissive Spatial Light Modulator. *Appl. Opt.* 2016, **55**, 6095–6101.
- [4]. Wang H, Du J, Wang H, et al. Generation of Spin - dependent accelerating beam with geometric metasurface[J]. *Advanced Optical Materials*, 2019, 7(15): 1900552.
- [5]. Song X, Huang L, Sun L, et al. Near-field plasmonic beam engineering with complex amplitude modulation based on metasurface[J]. *Applied Physics Letters*, 2018, 112(7).
- [6]. Zhang, S. et al. Generation of achromatic auto-focusing airy beam for visible light by an all-dielectric metasurface. *J. Appl. Phys.* **131**, 043104 (2022).
- [7]. https://www.thorlabs.com/newgrouppage9.cfm?objectgroup_id=11424.
- [8]. Dholakia, K. & Čižmár, T. Shaping the future of manipulation. *Nat. Photonics.* **5**, 335-342 (2011).
- [9]. Vettenburg, T. et al. Light-sheet microscopy using an Airy beam. *Nat. Methods.* **11**, 541-544 (2014).
- [10]. Jia, S., Vaughan, J. C. & Zhuang, X. Isotropic three-dimensional super-resolution imaging with a self-bending point spread function. *Nat. Photonics.* **8**, 302-306 (2014).
- [11]. Panagiotopoulos, P., Papazoglou, D., Couairon, A. et al. Sharply autofocused ring-Airy beams transforming into non-linear intense light bullets. *Nat. Commun.* **4**, 2622 (2013).
- [12]. Zhao, Y. et al. Performance evaluation of underwater optical communications using spatial modes subjected to bubbles and obstructions. *Opt. Lett.* **42**, 4699-4702 (2017).
- [13]. Chu, X. Evolution of an Airy beam in turbulence. *Opt. Lett.* **36**, 2701-2703 (2011).

We look forward to your favorable reply. Thank you.

Yours sincerely,

Chao Shen

The School of Information Science and Technology,

Fudan University

2005 Songhu Road, Shanghai, China

REVIEWERS' COMMENTS

Reviewer #1 (Remarks to the Author):

I am grateful to the authors for their careful responses to my comments. They have done a good job during the revision. The innovation of this article has been explained clearly. In particular, the new experiment directly shows the communication ability of the proposed system. So, I recommend the publication in Nature Communications.

Reviewer #2 (Remarks to the Author):

Dear Authors,

I think that the already good first version has been further improved providing a very good final work, moreover, some aspects, effectively not perfectly explained in the first version, have been clarified surely in the new version, but even and mainly in the rebuttal letter.

For these reasons, I'll suggest to accept the paper.